# Extended-Spectrum β-Lactam Resistant *Klebsiella* *pneumoniae* and *Escherichia coli* in Wild European Hedgehogs (*Erinaceus europeus*) Living in Populated Areas

**DOI:** 10.3390/ani11102837

**Published:** 2021-09-28

**Authors:** Biel Garcias, Laia Aguirre, Chiara Seminati, Nerea Reyes, Alberto Allepuz, Elena Obón, Rafael A. Molina-Lopez, Laila Darwich

**Affiliations:** 1Departament de Sanitat i Anatomia Animal, Universitat Autònoma de Barcelona (UAB), 08193 Cerdanyola del Vallès, Spain; biel.garcias@uab.cat (B.G.); Laia.aguirre@uab.cat (L.A.); Chiara.seminati@uab.cat (C.S.); Nerea.reyes@sanikolas.org (N.R.); Alberto.allepuz@uab.cat (A.A.); 2IRTA, Centre de Recerca en Sanitat Animal (CReSA, IRTA-UAB), Campus de la Universitat Autònoma de Barcelona, 08193 Cerdanyola del Vallès, Spain; 3Catalan Wildlife Service, Centre de Fauna Salvatge de Torreferrussa, 08130 Santa Perpètua de Mogoda, Spain; Elena.obon@gencat.cat (E.O.); Rafael.molina@gencat.cat (R.A.M.-L.)

**Keywords:** wild European hedgehogs, antimicrobial resistance, ESBL, *K. pneumoniae*, *E. coli*

## Abstract

**Simple Summary:**

The alarming emergence of antimicrobial resistance (AMR) in human and veterinary medicine has activated awareness for monitoring the levels of AMR pollution in the environment and wildlife. European hedgehogs (*Erinaceus europaeus*) are common wild species habiting urban areas in Europe. In this study, the occurrence and distribution of extended-spectrum β-lactam (ESBL) resistant enterobacteria and AMR genes were assessed in wild European hedgehogs in Catalonia, NE Spain. The results showed that 36.8% of the animals were detected as carriers of β-lactamase/carbapenemase resistance genes, with a special occurrence of human nosocomial bacteria such as *Klebsiella pneumoniae*, *Escherichia coli*, and *Citrobacter freundii*. In addition, more than half of the enterobacteria presented a multidrug resistance (MDR) phenotype and 31% of the isolates had an extended XDR profile. No differences in the spatial distribution of animals with AMR genes were observed within the study region. The results of this study suggest that the close contact with human areas predispose the transmission of AMR genes to wild hedgehogs because they either inhabit and/or feed in an anthropogenic environment. In conclusion, hedgehogs could be good sentinels or bioindicators of AMR environmental pollution, especially in highly populated areas with high human activity.

**Abstract:**

Wildlife has been suggested to be a good sentinel of environmental health because of its close interaction with human populations, domestic animals, and natural ecosystems. The alarming emergence of antimicrobial resistance (AMR) in human and veterinary medicine has activated/triggered the awareness of monitoring the levels of AMR pollution in wildlife. European hedgehogs (*Erinaceus europaeus*) are common wild species habiting urban areas in Europe. However, there are few studies conducted in hedgehogs as reservoirs of AMR bacteria or genes. The aim of this study was to assess the occurrence and distribution of ESBL, AmpC, and carbapenem-resistant enterobacteria and AMR genes in wild European hedgehogs in Catalonia, a densely populated region of NE Spain. A total of 115 hedgehogs admitted at the Wildlife Rehabilitation Center of Torreferrussa were studied. To our knowledge, this is the first description of β-lactam resistant enterobacteria in wild hedgehogs. Interestingly, 36.8% (42/114) of the animals were detected as carriers of β-lactamase/carbapenemase resistance genes. *Klebsiella* spp. (59.6%), and specifically *K. pneumoniae* (84.6%), were the bacteria with the highest proportion of resistance genes, followed by *E. coli* (34.6%) and *C. freundii* (5.8%). The most frequently detected genetic variants were *bla*CTX-M-15 (19.3%), *bla*SHV-28 (10.5%), *bla*CMY-1 (9.7%), *bla*CMY-2 (8.8%), and *bla*OXA-48 (1.7%). In addition, 52% (27/52) of the isolates presented a multidrug resistance (MDR) phenotype and 31% had an extended drug resistance (XDR) profile. No clustering of animals with AMR genes within the study region was shown in the spatial analysis, nor differences in the proportion of positive animals among regions, were detected. The results of this study suggest that wild European hedgehogs could be good sentinels of AMR environmental pollution, especially in areas with a high human population density, because they either inhabit and/or feed in an anthropogenic environment. In conclusion, it is crucial to raise awareness of the strong interconnection between habitats and compartments, and therefore this implies that AMR issues must be tackled under the One Health approach.

## 1. Introduction

Antimicrobial resistance (AMR) is a problem in human and veterinary medicine worldwide. The increasing prevalence of AMR in clinically important and commensal bacteria in both humans and livestock is attributed largely to selection through the use of antimicrobials [1,2]. However, the emergence of genes conferring resistance to human last-resort antibiotics, such as extended-spectrum β-lactamases (ESBLs), AmpC-type β-lactamases (AmpC), and carbapenem resistance genes, has been reported in the commensal bacteria of wildlife without any history of antimicrobial intake [3,4]. In fact, wild animals do not naturally come into contact with antimicrobials, but commensal bacteria have the potential to act as reservoirs of resistance genes contributing to the development of AMR in pathogens through horizontal transmission [3,5]. Thus, wildlife may be infected with antimicrobial-resistant bacteria (AMRB) or acquire AMR genes from human sources, agricultural facilities, and associated contaminated environments [6,7].

The European hedgehog (*Erinaceus europaeus*) is a common and widely distributed wild species in Europe. Hedgehogs are small, nocturnal, spiny-coated insectivores that live in a variety of habitats, generally in close contact with humans, but also with livestock in the countryside [8]. In recent years, hedgehog populations are increasingly inhabiting areas with human activity, such as gardens in residential areas or green areas in big cities [9]. Some of the consequences of this close hedgehog−human interaction is that people usually feed these animals and thus increases the risk of human physical contact. Moreover, the presence of hedgehogs in other anthropogenic environments, such as farms or crop fields, also allow for contact with livestock.

The prevalence of resistant bacteria in wildlife is linked with human activity [6]. Only a few studies have been conducted in hedgehogs reporting the presence of zoonotic antimicrobial resistant bacteria or AMR genes, such as β-lactam resistant *Enterobacteriaceae* in NE Spain [10], tetracycline resistance genes [11], or methicillin-resistant *Staphylococcus aureus* carrying *mecC* in Denmark and Sweden [12,13]. However, there is no research on the role of hedgehogs in AMRB dynamics at the interface between human populations, domestic animals, and natural ecosystems. This information is critical for implementing strategies to prevent further development and spread of AMRB between animals and humans.

Catalonia (NE Spain) is a densely populated county with a high density of pig fattening farms. Barcelona is Catalonia’s capital, one of Europe’s most densely populated cities. It is estimated that more than five million people live in the metropolitan area of Barcelona and the adjacent suburban areas, which is the largest on the Mediterranean Sea (Figure 1). Thus, the pressure on urban land on the main metropolitan areas of Barcelona is very high. As hedgehogs are a species closely related with the urban life, they could serve as reservoirs, vectors, and bioindicators of resistant bacterial pathogens and genetic determinants of AMR in the environment [6,7,14]. Therefore, the aim of this study was to assess the occurrence and distribution of ESBL, AmpC, and carbapenem-resistant enterobacteria in a wild population of European hedgehogs in Catalonia, NE Spain.

## 2. Materials and Methods

### 2.1. Study Population

From 2015 to 2019, faeces from 114 hedgehogs were collected at the Wildlife Rehabilitation Center (WRC) of Torreferrussa (Catalunya, North-East Iberian Peninsula). Sampling methods and animal handling techniques were in agreement with the Catalan Wildlife Service, which specifies the management protocols and the Ethical Principles according to Spanish legislation [15]. 

Upon admission, a physical exam under inhalant anaesthesia with isofluorane was performed on the animals and rectal samples were collected using sterile swabs with an Amies transport medium (Deltalab, Barcelona, Spain), before any pharmacologic or antimicrobial treatment. Data of the origin of the hedgehogs were provided by the wildlife rangers who brought the animals to the WRC. The age and sex of the animals were recorded by veterinary staff of the WRC. The cause of admission was related to the following categories [16]: orphaned young (27%), fortuitous finds (27%), metabolic/nutritional disease (15%), traumatic injuries (14%), natural disease/incident (13%), and other casualties such as illegal captivity or traps (4%).

### 2.2. Microbiological Analysis 

Rectal swabs were transported on an Amies medium to the Veterinary Infectious Diseases Diagnostic Laboratory located at the Autonomous University of Barcelona, Spain. Upon arrival, samples were processed immediately or were stored at 4 °C until being processed the next day. All rectal swabs were directly cultured on Columbia blood agar (BD GmBh, Munich, Germany) and MacConkey agar (Oxoid, Basingstoke, UK), not supplemented and supplemented with ceftriaxone (1 mg/L), and aerobically incubated for 24 h at 37 °C. The agar without ceftriaxone supplementation was used as a quality control of the sample growth to assess if negative growths on the supplemented agar were due to the ceftriaxone activity or because of a bad quality or degradation of the sample. All of the colonies grown on MacConkey agar with ceftriaxone were re-cultured on TSA agar in order to conduct bacterial identification, which was performed using conventional biochemical tests (oxidase, catalase, TSI, SIM, urease, citrate, and methyl red) and the API system (bioMérieux, Marcy l’Etoile, France).

### 2.3. Antimicrobial Susceptibility Test

All of the isolates identified from the MacConkey agar with ceftriaxone were stored at −80 °C before antimicrobial susceptibility testing was performed using the disk diffusion method [17]. Briefly, one to four colonies were suspended in 5 mL of distilled sterile water to achieve a turbidity of 0.5 in the McFarland scale. The dilution was then seeded onto Mueller–Hinton (Oxoid, Basingstoke, UK) plates. Each isolate was tested for the following antimicrobial groups, using commercial disks: aminopenicillins (ampicillin (BD) and amoxicillin/clavulanic acid (Oxoid)), cephalosporins (ceftiofur (BD), ceftriaxone (BD), and cefquinome (CondaLab)), fluoroquinolones (ciprofloxacin (BD) and enrofloxacin (BD)), aminoglycosides (gentamicin (BD)), tetracyclines (tetracycline (BD)), macrolides (erythromycin (BD)), lincospectin (Oxoid), sulfamethoxazole/trimethoprim (BD), and chloramphenicol (BD). Then, based on the laboratory testing readings, isolates were classified as susceptible, intermediate, or resistant, as previously described [18,19,20]. 

For statistical assessments, all isolates that exhibited intermediate resistance were re-classified as resistant. In addition, multidrug resistance (MDR) was defined as resistance to at least one agent in three or more antimicrobial categories; extensive drug resistance (XDR) was defined as resistance to all but two of the tested antimicrobial categories; and finally, pan-drug resistance (PDR) was defined as resistance to all of the categories tested [21].

### 2.4. Molecular Characterization of Antimicrobial Resistance Genes

DNA was extracted by boiling the bacterial growth from the selective MacConkey Agar supplemented with ceftriaxone (1 mg/L) plates, as previously described [18,22]. Briefly, all the bacterial growths were diluted in 600 μL of sterile distilled water, and 200 μL of the dilution was then transferred to a new tube. Two-hundred microliters of sterile distilled water was added to each tube. The tubes were boiled at 100 °C in a water bath for 10 min, and then centrifuged at 16,000× *g* for 5 min. After centrifugation, the supernatant was recovered and stored at −80 °C until processing.

β-lactamase—*bla*_TEM,_
*bla*_CTX,_
*bla*_SHV_, *bla*_CMY-1,_
*bla*_CMY-2_, and *bla*_OXA_—and plasmid mediated colistin—*mcr1*, *mcr2*, *mcr3*, *mcr4,* and *mcr5—*genes were assessed using previously described PCR protocols [10]. For all PCR assays, the master mix consisted of: 1 × PCR Buffer, 0.2 mM of each dNTP (Bioline, Livron-sur-Drôme, France), 3 mM of MgCl2, 1 mM of each primer, and 1 U of Taq DNA Polymerase (Bioline, Livron-sur-Drôme, France). A final volume of 2.5 μL of DNA was used in the PCR. In each reaction, positive and negative controls were included. The oligonucleotide sequence of primers is presented in Appendix A. Briefly, the PCR conditions were homogenized for all of the reactions as follows: 5 min at 94 °C, followed by 25 cycles of 1 min at 94 °C, 1 min of annealing at 55 °C and 1 min of extension at 72 °C, and a final extension step of 7 min at 72 °C. The controls for all genes were provided by the Veterinary Diagnostic Laboratory in Infectious Diseases of the Universitat Autònoma de Barcelona (UAB, Bellaterra, Spain).

Sanger DNA sequencing was conducted for *bla*_TEM,_
*bla*_CTX,_
*bla*_SHV_, and *bla*_OXA_ PCR products at the Genomic and Bioinformatics Service of the Universitat Autònoma de Barcelona (Spain). Sequences and chromatograms were manually explored to trim bad-quality bases with BioEdit 7.2. Once the assembly of the consensus sequences was conducted, both complete and partial sequences were aligned using the Clustal Omega program, and were finally *bla*sted against the public database (National Center for Biotechnology Information, NCBI, Bethesda, MD, USA).

### 2.5. Statistical Analysis

All of the statistical analyses were performed with R software, version 3.6.0 [23]. The graphics for representing antimicrobial susceptibility were built using the “AMR” package [24]. Generalised lineal models were used to test the association between the presence of AMR genes (considered as the dependent variable) and different variables such as age, sex, month, population, density of the municipality, and cause of admission to WRC (considered as explanatory variables). First, bivariate analysis was used to individually test all the variables, and those with a *p*-value lower than 0.3 were selected. After that, correlation among predictor variables was evaluated by calculating the Spearman correlation coefficient, and variables with a coefficient higher than 0.5 were removed. Non-correlated variables were included in the multivariate analysis and the best model was selected based on the lowest Akaike Information Criteria (AIC) value [25]. The limit of significance of the *p*-value was set at 0.05. 

Maps were created using Quantum Gis software, version 3.18 [26]. To evaluate the areas with a higher probability of having animals carriers of AMR genes, cluster analysis was performed with SaTScan version 9.6, using a purely spatial scan analysis with a Bernoulli model [27]. The spatial scan analysis is based in the likelihood ratio statistic based on the number of observed and expected cases in the specific zone and the search for clusters using a variable circular window size to detect spatial clusters in large areas while controlling for the underlying population [27]. The number of Montecarlo simulations was set at 999 and *p*-values lower than 0.05 were considered statistically significant. 

## 3. Results

Sixty seven out of the 114 wild hedgehog fecal samples analyzed (58.3%) had a positive growth on agar supplemented with ceftriaxone. Pure cultures were obtained from most of the animals and 19 out of 67 hedgehogs presented more than one colony, identifying a total of 90 bacterial isolates. *Klebsiella* spp., principally *Klebsiella pneumoniae* (29% out of the total bacteria), was the most frequently detected microorganism (42.2%), followed by *E. coli* (38.9%). Other bacteria such as *Citrobacter freundii* (5.6%), *Serratia* spp. (4.4%), *Proteus mirabilis* (3.3%), *Enterobacter cloacae* (2.2%), and *Shigella* spp. (1.1%) were also identified. 

Regarding the presence of β-lactamase resistance, 36.8% of the animals (42/114) were detected as carriers of AMR genes. *Klebsiella* spp. (59.6%) was the agent with the highest proportion of resistance genes, followed by *E. coli* (34.6%) and *C. freundii* (5.8%). The frequency of genes and variants in the studied population was as follows: 19.3% *bla*_CTX-M-15_, 10.5% *bla*_SHV-28_, 9.7% *bla*_CMY-1_, and 8.8% *bla*_CMY-2_ (Table 1). In addition, 1.7% of carbapenemases *bla*_OXA-48_ was found in isolates of *K. pneumoniae* and *E. coli* from two different animals (Table 2). In contrast, no colistin mcr-plasmid-mediated resistance genes were detected in the tested animals.

Regarding the AST results, almost 52% (27/52) of the isolates showed an MDR phenotype and 30.8% had an extended (XDR) profile (Table 2). Interestingly, a high proportion of the bacterial isolates presented more than one AMR gene, especially within the genus *Klebsiella*, in which 67.7% (21/31) of the isolates presented two or more β-lactamase genes. The most common combination was *bla*_CTX-M-15_ and *bla*_SHV-28,_ which was present in 21.2 % (11/52) of the overall isolates, and principally in *K. pneumoniae*, for which 84.6% of the isolates carried AMR genes, with up to 4–5 genes in some cases (Table 2).

AMP—ampicillin; AMC—amoxicillin + clavulanic acid; CEFQ—cefquinome; ENO—enrofloxacin; GM—gentamicin; E—erythromycin; LS—lincospectin; SXT—trimetoprim + sulphametoxazol; TET, Tetracycline; CIP—ciprofloxacin; CRO—ceftriaxone; XNL—ceftiofur; C—cloramphenicol; nt—not tested.

The results of the phenotypic antimicrobial susceptibility testing showed high levels of resistance in most of the tested antimicrobials (Figure 2). It is important to note the highest levels of resistance were observed among the *Klebsiella* isolates, especially for quinolones, third and fourth generation cephalosporines, trimethoprim/sulphametoxazol, and tetracyclines.

A generalised linear model was created to test the association between the presence of AMR genes and different variables, such as cause of hospitalization, human density of the zone, sex, age, and month of collection of the sample, but no variable was found to be statistically significant. As some animals were carriers from more than one bacterium, species could not be included as an explanatory variable. So, to evaluate the possible association of the variables explained before and the presence of the most prevalent genus, two specific generalised linear model were created just for animals that were carriers of *Klebsiella* spp and *E. coli*. However, again, no variable was found to be statistically significant. 

Odd ratios and *p*-values of the models are presented on Appendix A. Regarding animal distribution, at least one animal from the sampled regions was positive for β-lactams resistance genes (Figure 3). 

In Figure 4, a map with the location of wild hedgehogs positive for AMR genes is presented. Most of the sampled animals were found in highly populated areas of Catalonia. The scan statistics did not detect any significant cluster of animals carrying AMR genes, neither were there statistically significant differences in the frequency of positive animals among the regions.

## 4. Discussion

In this study, a high occurrence of β-lactamase resistance genes has been reported in a wild population of European hedgehogs coming from urbanized areas of Catalonia. To our knowledge, this is the first description of β-lactam resistant enterobacteria in wild hedgehogs. Sixty seven out of 114 faecal samples (57.8%) had growths in ceftriaxone supplemented agar, but AMR genes were detected in 42 of them (36.8%). Regarding the bacterial isolates, a total of 90 bacterial isolates were identified from those 67 analysed animals, and AMR genes were detected in 52 isolates. Thus, in line with other studies [28,29], we observed that some of the AMRB did not have any of the AMR genes. This could be explained by the possible existence of other mechanisms of resistance such as chromosomal mechanisms that were not included as a target of this study.

Interestingly, 36.8% of the animals were detected as carriers of AMR genes, with a special occurrence of human nosocomial bacteria harbouring ESBL, AmpC, and OXA genes. Moreover, *Klebsiella pneumoniae* was the bacteria with the highest proportion of resistance genes, followed by *E. coli* and *C. freundii.* The most frequently detected genes were *bla*_CTX-M-15_ (19.3%), *bla*_SHV-28_ (10.5%), *bla*_CMY-1_ (9.7%), and *bla*_CMY-2_ (8.8%). In addition, carbapenemases *bla*_OXA-48_ genes were found in two animals. It is interesting to remark that more than half of the enterobacteria presented a MDR phenotype, with an extended profile (XDR) in 30% of the isolates.

According to the World Health Organization and the World Organization for Animal Health, third- and fourth-generation cephalosporins are critically important antibiotics to treat infection diseases in humans [30] and animals [31]. However, in the last decades, acquired resistance to β-lactams, mainly mediated by extended-spectrum beta-lactamase and AmpC β-lactamases, such as cephalosporinases, has been widely reported worldwide from humans, livestock, companion animals, the environment, and wildlife [32,33]. The results obtained in this study in wild hedgehogs dwelling in highly populated areas of Catalonia corroborate this wide distribution of β-lactam resistance genes. In fact, a high proportion of the bacterial isolates, especially *K. pneumoniae*, presented two or more β-lactamase genes and a phenotype of resistance to third- and fourth-generation cephalosporins among other compounds.

The presence of β-lactamase resistance genes has been historically analysed using *E. coli* as a marker in livestock [34] and wildlife [35,36,37]. Regarding wildlife, the presence of ESBL genes has been reported principally in wild birds [38,39,40], but also in other species such as bats [41], wild boars [29], and fish [42]. The most commonly detected ESBL genes are CTX-M-1, CTX-M-14, and CTX-M-15, which are also found frequently in hospital settings, suggesting a relationship between bacteria from wildlife and health-care facilities [33]. Interestingly, hedgehogs from this study harbouring the CTX-M group were all classified with the CTX-M-15 variant. This variant has been described in the high risk clone ST131 of *E. coli*, which generates nosocomial issues worldwide [43]. Within the SHV group, we found a variety of variants distributed in all of the regions sampled, with SHV-28 being the most frequent one. The combination of CTX-M-15 with SHV-28 was the most repeated association, found principally in *K. pneumoniae* isolates. This association, when located in *Klebsiella* species, has been related with the high risk clone ST15, which is an emerging cause of nosocomial infections [44].

The AmpC β-lactamase genes CMY-1 and CMY-2 were detected in the tested hedgehogs. As these genes can be either acquired by plasmids or can be chromosomally selected [44], it is difficult to attribute the presence of CMY-1 and -2 in these animals as a result of direct environmental contamination. Similarly, the genetic resistance of *C. freundii* isolates found in three animals is usually originated at a chromosomal level. However, most of the bacteria carrying AmpC genes found in the wild hedgehogs of this study were *Klebsiella* spp. and *E. coli*, which usually acquire resistance genes by plasmids [45].

Regarding other families of antibiotics, plasmid mediated colistin resistance genes were also analysed. Colistin is considered a last therapeutic option in human medicine and is widely used in cases of infections caused by ESBL bacteria [46]. However, the coexistence of resistance genes to β-lactamase and to colistin has been detected in humans [47], livestock [48], and wild birds [49]. It has been speculated that colistin use in animal production, and particularly in the pig industry, could be the original source of these genes [22]. In this study, none of the tested animals presented colistin (mcr-1 to -5) resistance genes, but this is an expected finding as most of the tested animals came from urbanized areas with a low risk of contact with residues from pig production. However, the inclusion of a larger sample size with hedgehogs from regions with a high swine population density, such as Osona, in the analysis, could help to improve the odds of detecting these colistin resistance genes. On the other hand, the emergence of resistance to carbapenems—last resort drugs used in hospital settings for the treatment of severe infections caused by β-lactam resistant enterobacteria—has compromised the therapeutic options in human health [50]. Specially, the OXA-48 variant of carbapenemases is becoming highly prevalent in human clinical infections in Europe [51]. This high prevalence in human settings could explain the 1.7% of OXA-48 frequency in the hedgehog populations of urban surroundings. As carbapenems are not allowed in food animals, the increasing presence of carbapenemases genes in livestock [52], companion animals [53], and wildlife animals [54] is alarming evidence of the anthropogenic environmental contamination.

The spatial analysis showed that the frequency and distribution of hedgehogs carrying AMR genes was similar in the study area. However, it is important to mention that most of the animals included in this study came from highly urbanized regions, because there is more chance of them being found and taken to the wildlife centre. In addition, the hedgehog population density has increased in urban areas compared with rural zones (26), so sampling animals from these zones can be more difficult. Finally, as most of the animals came from highly populated regions, it is reasonable to speculate that the close contact with human areas leads to the transmission of AMR genes to wildlife, because they either inhabit and/or feed in an anthropogenic environment [55].

Regarding the AMR phenotype, it is important to remark that, although the AST was only performed on isolates grown on MacConkey agar supplemented with ceftriaxone, in order to focus on ESBLs, some of these strains showed a sensitive phenotype to ceftriaxone after long storage at −80 °C. These long periods of freezing could affect both the viability of some of the strains and the loss of these genes that confer resistance.

The high frequency of hedgehogs carrying β-lactamase resistance genes in this study is a cause of concern, because these animals have never been treated with an antibiotic; thus, the presence of AMR genes must be related to exposure to sewage water or soils contaminated with AMR bacteria and/or antimicrobial residues. Sewage treatment plants are not completely efficient at removing all antibiotic residues and AMR bacteria [56], and contaminated water is spread into rivers and seas. Moreover, livestock manure is applied to land and can also contain it [57], enriching the resistome in both soils and earthworms [58]. Earthworms, a key biological component of soil, have been shown to contribute to spreading AMR, as they can acquire antibiotics directly from the soil [55]. Considering that most of the hedgehog’s diet is based on earthworms and other invertebrates, hedgehogs are more directly exposed to these antibiotic substances and resistance genes. Hence, the results of this study suggest that hedgehogs are good sentinels of AMR environmental pollution, especially in highly populated areas with high human activity.

Furthermore, the presence of wild hedgehogs carrying MDR bacteria as clinical patients at the WRC might compromise the effectiveness of antimicrobial treatments in these animals, and it can also represent a zoonotic risk for staff and the general population who come into contact with them.

## 5. Conclusions

This study reveals a high prevalence of human nosocomial-like bacteria with β-lactamase resistance genes in wild hedgehogs inhabiting highly populated zones in Catalonia. It is crucial to raise awareness about the strong interconnection between habitats and compartments induced by multiple exchange routes, which implies that AMR issues must be tackled under the One Health approach. Further studies are needed to confirm the relatedness of cephalosporin resistant *Enterobacteriaceae* strains found in wild hedgehogs as well as those that cause concern in human hospitals at a regional level.

## Figures and Tables

**Figure 1 animals-11-02837-f001:**
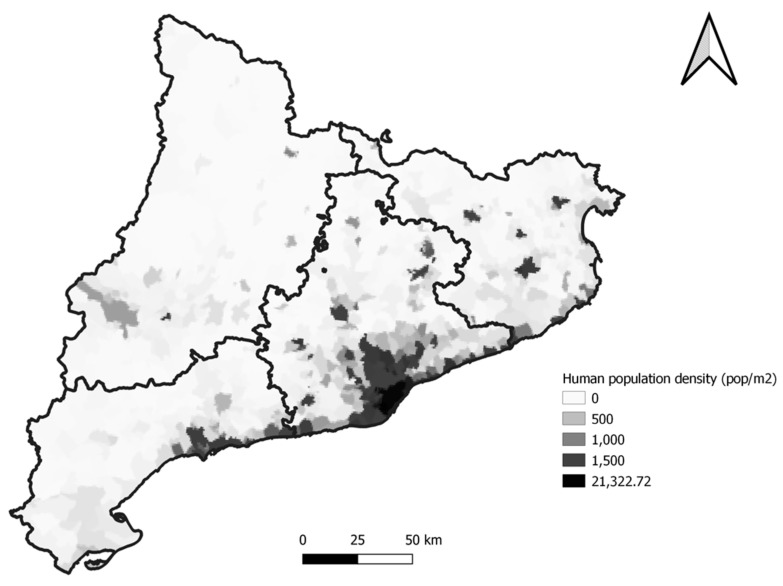
The map shows the human population density of Catalonia. Sampled animals were concentrated in the zones with a higher density.

**Figure 2 animals-11-02837-f002:**
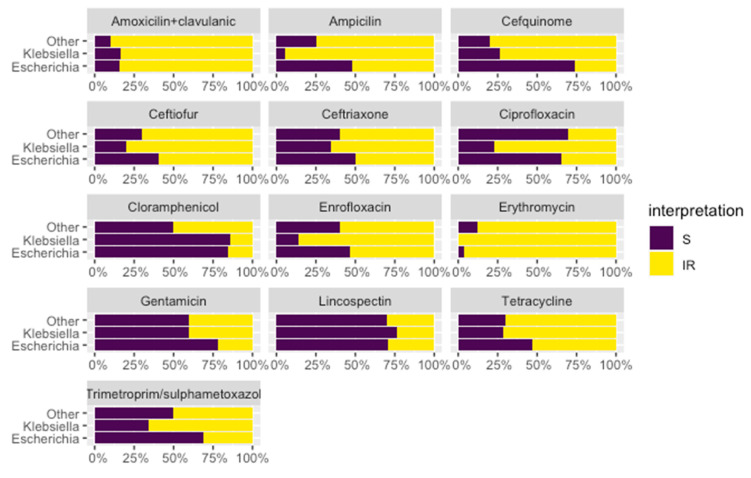
Antimicrobial susceptibility phenotype of the different isolates. Interpretation: S—sensitive (in purple); I—intermediate and R—resistant (in yellow).

**Figure 3 animals-11-02837-f003:**
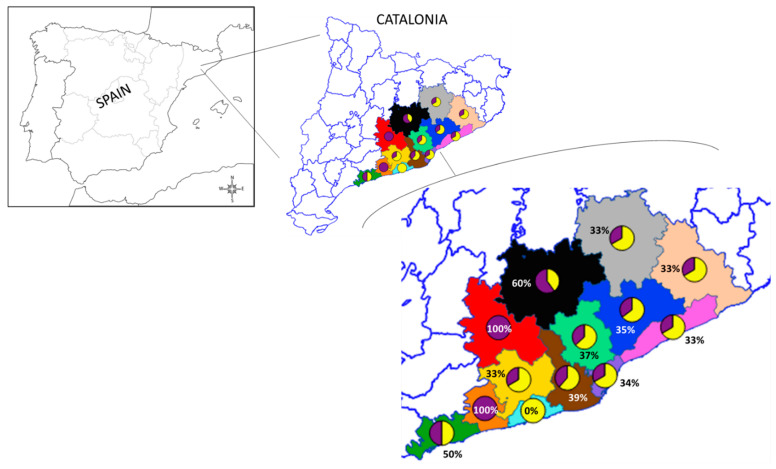
The map shows the counties where hedgehogs were found. Pie charts represent the occurrence of wild hedgehogs positive for AMR genes from the total number of sampled individuals for the corresponding study areas: green—Tarragonès (*n* = 2); orange—Baix Penedès (*n* = 1); azure—Garraf (*n* = 2); yellow—Alt Penedès (*n* = 3); brown—Baix Llobregat (*n* = 13); red—Anoia (*n* = 1); Black—Bages (*n* = 5); mint—Vallès Occidental (*n* = 38); purple—Barcelonès (*n* = 17); blue—Vallès Oriental (*n* = 17); pink—Maresme (*n* = 6); beige—Selva (*n* = 3); grey—Osona (*n* = 3). Pie charts: purple—positive; yellow—negative.

**Figure 4 animals-11-02837-f004:**
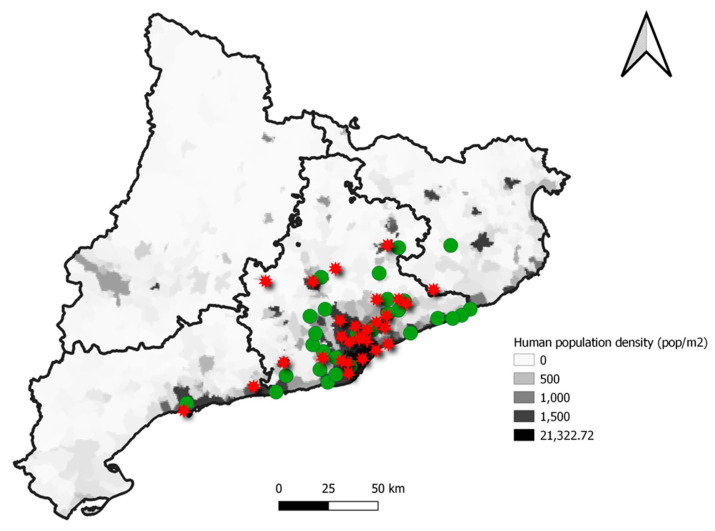
Localization and distribution of the hedgehogs included in the study according to the presence of at least one AMR gene (red star) or their absence (green dot) in each municipality over the human population density of each zone.

**Table 1 animals-11-02837-t001:** Number and frequencies of the samples positive for β-lactamase genes in European hedgehogs. CI, confidence intevals.

AMR Gene	Number of Animals with AMR Gene	Overall Frequency (%) CI 95% (*n* = 114)	Frequency (%) and CI 95% among Postive Animals (*n* = 42)
*bla* _CTX-M-15_	22	19.3 (12.1–26.5)	52.4 (37.3–67.5)
*bla* _CTX-M-3_	1	0.9 (0–2.6)	2.4 (0–7)
*bla* _SHV-1_	5	4.4 (0.6–8.2)	11.9 (2.1–21.7)
*bla* _SHV-11_	5	4.4 (0.6–8.2)	11.9 (2.1–21.7)
*bla* _SHV-12_	3	2.6 (0–5.6)	7.1 (0–14.9)
*bla* _SHV-28_	12	10.5 (4.9–16.2)	28.6 (14.9–42.2)
*bla* _TEM-1b_	5	4.4 (0.6–8.2)	11.9 (2.1–21.7)
*bla* _CMY-1_	11	9.7 (4.2–15.1)	26.2 (12.9–39.5)
*bla* _CMY-2_	10	8.8 (3.6–14)	23.8 (10.9–36.7)
*bla* _OXA-48_	2	1.8 (0–4.2)	4.8 (0–11.2)

**Table 2 animals-11-02837-t002:** Antimicrobial resistance genotypes and phenotypes of β-lactamase producing *Enterobacteriaceae* detected in wild European hedgehogs.

*Bacterial* spp.	*β-lactam* Resistance Genes	AMR Phenotyping
** *Escherichia coli* **	CTX-M-15, SHV-28	AMC, E, TET
	CTX-M-15	CEFQ, E, CRO, XNL
	CTX-M-15	E
	CTX-M-15	AMP, CEFQ, ENO, E, LS, SXT, GM, CIP, CRO, C, XNL
	CTX-M-15	nt
	CTX-M-15	E, GM
	CTX-M-15	AMP, CEFQ, ENO, E, TET, CIP, CRO, XNL
	CTX-M-15	AMP, AMC, CEFQ, ENO, GM, E, SXT, TET, CIP, CRO, XNL
	SHV-12	AMP, AMC, CEFQ, ENO, GM, E, SXT, TET, CIP, CRO, XNL
	SHV-11, OXA-48	AMP, AMC, CEFQ, E, CRO, XNL
	TEM-1b, CMY-2	AMP, AMC, E, SXT, CRO, XNL
	CMY-2	AMP, ENO, CIP, CRO, XNL
	CMY-2	AMP, SXT, TET, CRO, XNL
	CMY-2	AMP, CRO, XNL
	CMY-2	AMP, ENO, E, LS, TET, XNL
	CMY-2	AMP, AMC, ENO, E, LS, SXT, TET, CIP, CRO, XNL
	CMY-2	AMP, AMC, E, CRO, XNL
	CMY-2	AMP, AMC, E, TET
** *Klebsiella* ** **spp.**	SHV-28, TEM-1b, CMY-2	AMP, AMC, ENO, GM, E, LS, SXT, TET, CIP, C
	SHV, CMY-1	AMP, ENO, E, LS, SXT, TET, CIP, C
	CTX-M-15, SHV-1	AMP, AMC, CEFQ, ENO, E, SXT, TET, CIP, CRO, XNL
	CMY-2	AMP, AMC, CEFQ, ENO, E, SXT, TET, CIP, CRO, C, XNL
	CTX-M-15, SHV-28, CMY-1	AMP, AMC, CEFQ, ENO, GM, E, SXT, TET, CIP, CRO, XNL
	CTX-M-15, SHV-11, CMY-1	AMP, AMC, CEFQ, ENO, E, SXT, TET, CIP, CRO, XNL
	CTX-M-15, SHV-11, CMY-1	AMP, AMC, CEFQ, ENO, E, SXT, TET, CIP, CRO, XNL
	CTX-M-15, SHV-11	AMP, CEFQ, ENO, E, TET, CIP, CRO, XNL
	CTX-M-15	AMP, AMC, CEFQ, ENO, E, SXT, TET, CIP, CRO, XNL

** *K. pneumoniae* **	CTX-M-15, CMY-1	AMP, AMC, CEFQ, ENO, E, SXT, TET, CIP, CRO, XNL
	CTX-M-15, SHV-28, CMY-1	AMP, AMC, CEFQ, ENO, GM, E, SXT, TET, CIP, CRO, XNL
	CTX-M-15, SHV-28, CMY-1	AMP, AMC, CEFQ, ENO, GM, E, SXT, TET, CIP, CRO, XNL
	CMY-1, OXA-48	AMP, AMC, ENO, E, LS, SXT, CIP, XNL
	SHV-28, CMY-1	AMP, AMC, ENO, E, LS, SXT, TET, CIP, XNL
	CTX-M-15	AMP, AMC, CEFQ, ENO, GM, E, SXT, TET, CIP, CRO, XNL
	CTX-M-15, SHV-28	AMP, AMC, CEFQ, ENO, GM, E, SXT, TET, CIP, CRO, XNL
	CTX-M-15, SHV-28	AMP, AMC, CEFQ, ENO, GM, E, LS, SXT, TET, CIP, CRO, XNL
	CTX-M-15, SHV-28	AMP, AMC, E
	CTX-M-15, SHV-28	AMP, AMC, ENO, E, TET, CRO, XNL
	CTX-M-15, SHV-28	nt
	SHV-1	AMP, AMC, CEFQ, ENO, E, SXT, TET, CIP, CRO, C, XNL
	SHV-1	AMP, AMC, ENO, E, CRO, XNL
	SHV-1	AMP, AMC, ENO, E, XNL
	CTX-M-15, SHV-28	AMP, AMC, CEFQ, ENO, GM, E, SXT, TET, CIP, CRO, XNL
	CTX-M-15, SHV-28, CMY-1, CMY-2	AMP, AMC, CEFQ, ENO, GM, E, SXT, CIP, CRO, XNL
	CTX-M-15, SHV-1, TEM-1b, CMY-1, CMY-2	AMP, AMC, CEFQ, ENO, GM, E, SXT, TET, CIP, CRO, XNL
	SHV-28	AMP, AMC, CEFQ, ENO, GM, E, SXT, TET, CIP, CRO, XNL
	SHV-12	AMP, AMC, CEFQ, ENO, E, SXT, TET, CIP, CRO, XNL
	SHV-1, TEM-1b	AMP, AMC, CEFQ, ENO, GM, E, SXT, TET, CIP, CRO, XNL
	SHV-1	nt

** *K. oxytoca* **	CTX-M-3	AMP, AMC, CEFQ, ENO, E, CIP, CRO, XNL

** *Citrobacter freundii* **	SHV-12, CMY-2	AMP, AMC, CEFQ, ENO, E, LS, SXT, TET, CIP, CRO, XNL
	TEM-1b, CMY-2	AMP, AMC, CEFQ, ENO, GM, E, TET, CIP, CRO, XNL
	CMY-2	nt

AMP—ampicillin; AMC—amoxicillin + clavulanic acid; CEFQ—cefquinome; ENO—enrofloxacin; GM—gentamicin; E—erythromycin; LS—lincospectin; SXT—trimetoprim + sulphametoxazol; TET, Tetracycline; CIP—ciprofloxacin; CRO—ceftriaxone; XNL—ceftiofur; C—cloramphenicol; nt—not tested.

## Data Availability

Data will be made available upon reasonable request to the corresponding author.

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
