# Peer review of "Extended-Spectrum β-Lactam Resistant *Klebsiella* *pneumoniae* and *Escherichia coli* in Wild European Hedgehogs (*Erinaceus europeus*) Living in Populated Areas"

_animals, 2021, doi:10.3390/ani11102837_

Round 1

Reviewer 1 Report

The manuscript is much improved, and I thank the authors for making the modifications which I suggested. There are a few more very minor changes which need to be made before acceptance.

Line 78- reword to environment, such as farms …..

Line 163- for all PCR assays (reword)

Line 203- pure cultures were obtained from most of the animals ….(Reword)

Line 246- as some animals were carriers … (Reword)

Line 247- as an explanatory variable (reword)

Line 249- a specific generalised …(reword)

Line 277- explained rather than explain

Line 324- with SV-28 being the most …(reword)

Author Response

The manuscript is much improved, and I thank the authors for making the modifications which I suggested. There are a few more very minor changes which need to be made before acceptance.
Thank you very much for your useful comments and criticism. Here you are the modifications introduced in the new version of the manuscript:
Line 78- reword to environment, such as farms …..

Done as follows:“Moreover, the presence of hedgehogs in other anthropogenic environment such as farms or crop fields also let the contact with livestock.”
Line 163- for all PCR assays (reword)

Done as follows: “For all PCR assays, the master mix consisted of…”

Line 203- pure cultures were obtained from most of the animals ….(Reword) Done: “Sixty seven out of the 114 wild hedgehog fecal samples analyzed (58.3%) had a positive growth on agar supplemented with ceftriaxone. Pure cultures were obtained from most of the animals and 19 out of 67 hedgehogs presented more than one colony, identifying a total of 90 bacterial isolates.”
Line 246- as some animals were carriers … (Reword)

Done;

Line 247- as an explanatory variable (reword)

Done; Line 249- a specific generalised …(reword). We changed the paragraph as follows:“As some animals were carriers from more than one bacterium, species could not be included as an explanatory variable. So, to evaluate the possible association of the variables explained before and the presence of the most prevalent genus, two specific generalised linear model were created just for animals that were carriers of Klebsiella spp and E. coli.”
Line 277- explained rather than explain

Done
Line 324- with SV-28 being the most …(reword)

Done: “Within the SHV group, we found a variety of variants distributed in all the regions sampled, with SHV-28 being the most frequent one.

Reviewer 2 Report

The manuscript "Extended-spectrum β-lactam resistant Klebsiella pneumoniae and Escherichia coli in wild European hedgehogs (Erinaceus europeus) living in populated areas" by Garcias et al. is significantly improved after the revision phase. However, I think that some comments included in the file of Authors’ responses to Reviewers’ comments should also be added in the manuscript so that the text could become even more clear. Please see the details below.

Major revisions

Materials and Methods

  • Figure 1: in the map legend it would be better including a scale of values instead of the two extremes (0 and 21323), it would be more clear.
  • Lines 113-114: a few sentences about this Authors’ comment “We decided to use agar without ceftriaxone supplementation to have a quality control of the sample growth. For instance, to assess negative growths on supplemented agar due to the effect of the ceftriaxone activity instead of a bad quality or degradation of the sample.” should be added in the manuscript in order to make the text more clear.
  • Lines 115-116: “Bacterial identification was just done on agar supplemented with ceftriaxone” this concept should be added also in the manuscript.
  • Lines 119-121: “Isolates grown on MacConkey agar with ceftriaxone were re-cultured on TSA agar and antimicrobial susceptibility test using Kirby-Bauer method was performed then.” this concept should be added also in the manuscript.

Results

  • Lines 176-177: according to the Authors’ comment I was wondering if for each sample that had more than one colony, did you test all the colonies in the plate? This explanation should be added in the manuscript.
  • Following the Authors’ response about “bacterial species” included as explanatory variable in the regression model, for completeness/correctness also a statistical analyses for coli should be included since prevalences of these two strains vary little, 42.2% and 38.9%.
  • Lines 204-207: “The antimicrobial susceptibility test (AST) were just performed on isolates grown on MacConkey agar + ceftriaxone, in order to focus on ESBLs. And this AMR genes were found on these isolates.” this concept should be added also in the manuscript.
  • Lines 204-207: “However, AST was done after quarantine, so that means that isolates were frozen at -80ºC and this could affect to the viability of some of the strains and lose some of the genes that confer resistance. However, we don’t have a certainty of that and we don’t have any other explanation.” this criticism/limitation of the study should be included in the text.

Author Response

The manuscript "Extended-spectrum β-lactam resistant Klebsiella pneumoniae and Escherichia coli in wild European hedgehogs (Erinaceus europeus) living in populated areas" by Garcias et al. is significantly improved after the revision phase. However, I think that some comments included in the file of Authors’ responses to Reviewers’ comments should also be added in the manuscript so that the text could become even more clear. Please see the details below.
Major revisions
Materials and Methods
Figure 1: in the map legend it would be better including a scale of values instead of the two extremes (0 and 21323), it would be more clear.

Done (see Fig 1 new R2 version)
Lines 113-114: a few sentences about this Authors’ comment “We decided to use agar without ceftriaxone supplementation to have a quality control of the sample growth. For instance, to assess negative growths on supplemented agar due to the effect of the ceftriaxone activity instead of a bad quality or degradation of the sample.” should be added in the manuscript in order to make the text more clear.

Done:” Agar without ceftriaxone supplementation was used as a quality control of the sample growth to assess if negative growths on supplemented agar were due to the ceftriaxone activity or a bad quality or degradation of the sample. All colonies grown on MacConkey agar with ceftriaxone were re-cultured on TSA agar in order to conduct bacterial identification, which was performed using conventional biochemical tests (oxidase, catalase, TSI, SIM, urease, citrate and methyl red) and the API system…”
Lines 115-116: “Bacterial identification was just done on agar supplemented with ceftriaxone” this concept should be added also in the manuscript. Lines 119-121: “Isolates grown on MacConkey agar with ceftriaxone were re-cultured on TSA agar and antimicrobial susceptibility test using Kirby-Bauer method was performed then.” this concept should be added also in the manuscript.

Done:“Isolates grown on MacConkey agar with ceftriaxone were re-cultured on TSA agar and antimicrobial susceptibility test was performed using the disk diffusion method [17].”
Results
Lines 176-177: according to the Authors’ comment I was wondering if for each sample that had more than one colony, did you test all the colonies in the plate? This explanation should be added in the manuscript.

Yes, we test all colonies. Done:“All colonies grown on MacConkey agar with ceftriaxone were re-cultured on TSA agar in order to conduct bacterial identification, which was performed using conventional biochemical tests…”
Following the Authors’ response about “bacterial species” included as explanatory variable in the regression model, for completeness/correctness also a statistical analyses for coli should be included since prevalences of these two strains vary little, 42.2% and 38.9%.

We have done the analysis just for E. coli, but again no variable appears to be significative. We have added in Results section.
Lines 204-207: “The antimicrobial susceptibility test (AST) were just performed on isolates grown on MacConkey agar + ceftriaxone, in order to focus on ESBLs. And this AMR genes were found on these isolates.” this concept should be added also in the manuscript. Lines 204-207: “However, AST was done after quarantine, so that means that isolates were frozen at -80ºC and this could affect to the viability of some of the strains and lose some of the genes that confer resistance. However, we don’t have a certainty of that and we don’t have any other explanation.” this criticism/limitation of the study should be included in the text.

The new version appears like:“The antimicrobial susceptibility test (AST) was just performed on isolates grown on MacConkey agar supplemented with ceftriaxone, in order to focus on ESBLs. And this AMR genes were found on these isolates. However, some of these strains showed a sensitive phenotype to ceftriaxone after long storage at -80ºC. These long periods of freezing could affect both the viability of some of the strains and the loose of these genes that confer resistance.”

Reviewer 3 Report

Please, upload the manuscript in a clear version since the current file contains the "track-change" . The tables are presented not in an unique outline and form. The results are poorly presented.

The Ethical Approval Number is not provided.

Could the authors please discuss how their findings impact the animals' life. Is the treatment of found wild animals very frequent or rare?

Author Response

Please, upload the manuscript in a clear version since the current file contains the "track-change" . The tables are presented not in an unique outline and form. The results are poorly presented.
Thank you very much for your comments. We followed the journal guidelines and all the new changes introduced form the first round revision, were keep it for the reviewers. However, in this R2 round we have tried to show only the principal changes required by the reviewers (underlined text). All changes due to English revision has not been tracked.
The Ethical Approval Number is not provided.
Ethical approval is not required for this kind of studies. Indeed, the manuscript includes the following sentence “Sampling methods and animal handling techniques were in agreement with the Catalan Wildlife Service who specify the management protocols and the Ethical Principles ac-cording to the Spanish legislation [15]”

Could the authors please discuss how their findings impact the animals' life. Is the treatment of found wild animals very frequent or rare?
Done. A last sentence has been included at the discussion section:“Furthermore, the presence of wild hedgehogs carrying MDR bacteria as clinical pa-tients at the WRC might compromise the effectiveness of antimicrobial treatments in these animals, as well as it can represent a zoonotic risk for staff and the general population in contact with them.”

Round 2

Reviewer 2 Report

I've just two last minor comments:

  • this sentence "The antimicrobial susceptibility test (AST) was just performed on isolates grown on MacConkey agar supplemented with ceftriaxone, in order to focus on ESBLs. However, some of these strains showed a sensitive phenotype to ceftriaxone after long storage at -80ºC. " should be moved from the Results paragraph to the Materials and Methods and explaining why, after sampling, only a few strains were stored at -80.
  • this sentence "These long periods of freezing could affect both the viability of some of the strains and the loose of these genes that confer resistance. " should be moved to the Discussion paragraph.

Author Response

  • This sentence "The antimicrobial susceptibility test (AST) was just performed on isolates grown on MacConkey agar supplemented with ceftriaxone, in order to focus on ESBLs. However, some of these strains showed a sensitive phenotype to ceftriaxone after long storage at -80ºC. " should be moved from the Results paragraph to the Materials and Methods and explaining why, after sampling, only a few strains were stored at -80.

Answer: I think there is a misunderstanding. All bacterial isolates were stored at -80ºC (It is the reasonable way to proceed in microbiology to preserve bacterial strains for further analysis). However, in this particular study it seems that some isolates have lost the resistance to ceftriaxone and one of our hypothesis is that long periods of freezing can alter the gene expression or the phenotypic sensitivity to this drug. In MM section, now it is better explained this issue: “All isolates identified from the MacConkey agar with ceftriaxone were stored at -80ºC before antimicrobial susceptibility testing was performed by the disk diffusion method [17]”.

  • This sentence "These long periods of freezing could affect both the viability of some of the strains and the loose of these genes that confer resistance. " should be moved to the Discussion paragraph.

      Answer: done. This paragraph has been moved to the discussion section as follows: 

“Regarding to AMR phenotype, it is important to remark that, although the AST was just performed isolates grown on MacConkey agar supplemented with ceftriaxone, in or-der to focus on ESBLs, some of these strains showed a sensitive phenotype to ceftriaxone after long storage at -80ºC. These long periods of freezing could affect both the viability of some of the strains and the loose of these genes that confer resistance.”

Reviewer 3 Report

I have no further comments!

Author Response

no comments

This manuscript is a resubmission of an earlier submission. The following is a list of the peer review reports and author responses from that submission.

Round 1

Reviewer 1 Report

This is a lovely, well executed, and novel study investigating ESBLs in wild hedgehogs in Spain. It is generally well written and well carried out and is a great addition to the literature. I have made a few minor comments below which are mainly grammatical issues, but these changes are very minor.

Please ensure that all bacterial names are put into italics throughout the manuscript.

Line 24- please define MDR and XDR in the abstract

Line 25- were rather than where

Line 41- maybe swap agent for bacteria, or pathogen?

Line 45- show rather than showed

Line 46- neither were differences detected (reword)

Line 62- they may encounter antimicrobial residues, such as in water sources?

Line 67- maybe ‘The European hedgehog’ may sound better?

Line 68- maybe habitats may sound better?

Line 74- this may read better as ‘and thus increases the risk of human physical contact’?

Line 76- this may read better as  ‘Only a few studies have been conducted in hedgehogs…’

Line 78- Staph aureus needs to be italicised

Line 79- Should MecC be italicised?

Line 82- you could use AMRB here again

Line 84- what does NE mean here?

Line 85- delete ‘the’

Line 86- maybe say five million people?

Line 96- ‘collected at the Wildlife…’ (reword)

Line 105-108- this may be better in the results- but am happy for you to leave it here if you prefer. May also be nice to include the n number here.

Line 115- please include antibiotic manufacturer

Line 115- replace during with ‘for’

Line 116- please detail these biochemical tests

Line 120- maybe state that these colonies were from the original plate. And how were they chosen?

Line 123-127- concentrations and manufacturers in here would be useful

Line 139- guessing at 100C but please state

Line 140- can you please add this speed in as xg?

Line 146- could you please add in the PCR reagents here?

Line 158- maybe with the AMR package?

Line 159- is this generalised linear models?

Line 177- up to 90 isolates? Or 90 isolates? It just sounds a bit vague given that you know how many you found

Line 187- please ensure that there is a space between the genus and the species name of the bacteria. This line in particular but please check throughout.

Table 1- please change gen to gene

Line 193- remarkable within the genus Klebsiella, in which  …. (reword)

Line 206- were observed …. (reword)

Line 206- comma after isolates

Line 211- maybe a generalised linear model?

Line 314- I would like to see some more details in here- perhaps a table of the p values for these?

Line 218- comma after genes

Line 218- neither were there … (reword)

Line 222- It would be good to have in here which colour means positive?

Line 237- maybe change agent for pathogen/ bacteria?

Line 239-240- do these gene names need to be italicised?

Line 261- please reword this as it is unclear what you are saying

Line 262- within the SHV group (reword)

Line 263- with SHV-28  being the most frequent one (reword)

Line 279- maybe have this as humans?

Line 285- you have enterobacteria with a capital and a non capital throughout, please check and be consistent

Line 291- delete an

Line 306- efficient at removing all … (reword)

Line 311- Considering that most of the hedgehog’s (reword)

Line 356- 258- please reword this as it is a bit unclear

Author Response

REVIEWER 1

This is a lovely, well executed, and novel study investigating ESBLs in wild hedgehogs in Spain. It is generally well written and well carried out and is a great addition to the literature. I have made a few minor comments below which are mainly grammatical issues, but these changes are very minor.

 Please ensure that all bacterial names are put into italics throughout the manuscript.

 Line 24- please define MDR and XDR in the abstract DONE

Line 25- were rather than where DONE

Line 41- maybe swap agent for bacteria, or pathogen? DONE

Line 45- show rather than showed DONE

Line 46- neither were differences detected (reword) DONE

 Line 62- they may encounter antimicrobial residues, such as in water sources? DONE

Line 67- maybe ‘The European hedgehog’ may sound better? DONE

Line 68- maybe habitats may sound better? DONE

Line 74- this may read better as ‘and thus increases the risk of human physical contact’? DONE

Line 76- this may read better as  ‘Only a few studies have been conducted in hedgehogs…’ DONE

Line 78- Staph aureus needs to be italicised DONE

Line 79- Should MecC be italicised? DONE

Line 82- you could use AMRB here again DONE

Line 84- what does NE mean here? DONE

Line 85- delete ‘the’ DONE

Line 86- maybe say five million people? DONE

Line 96- ‘collected at the Wildlife…’ (reword) DONE

Line 105-108- this may be better in the results- but am happy for you to leave it here if you prefer. May also be nice to include the n number here. DONE

Line 115- please include antibiotic manufacturer DONE

Line 115- replace during with ‘for’ DONE

Line 116- please detail these biochemical tests DONE

Line 120- maybe state that these colonies were from the original plate. And how were they chosen? Usually, we have pure cultures, and we collect the representative colony. However, 19 animals have two colonies that were growing with the same magnitude and we collected both. Now this has been included in the results as follows:

“Sixty seven out of 114 wild hedgehogs analyzed (58.3%) had a positive growth on agar supplemented with ceftriaxone. Most of the animals presented pure cultures and 19 out of 67 hedgehogs presented more than one colony, identifying a total of 90 bacterial isolates.”

Line 123-127- concentrations and manufacturers in here would be useful.

 We would like to add it as supplementary table S1

Line 139- guessing at 100C but please state DONE

Line 140- can you please add this speed in as xg? DONE

Line 146- could you please add in the PCR reagents here? DONE

Line 158- maybe with the AMR package? DONE

Line 159- is this generalised linear models? DONE

Line 177- up to 90 isolates? Or 90 isolates? It just sounds a bit vague given that you know how many you found DONE

Line 187- please ensure that there is a space between the genus and the species name of the bacteria. This line in particular but please check throughout. DONE

Table 1- please change gen to gene DONE

Line 193- remarkable within the genus Klebsiella, in which  …. (reword) DONE

Line 206- were observed …. (reword) DONE

Line 206- comma after isolates DONE

Line 211- maybe a generalised linear model? DONE

Line 314- I would like to see some more details in here- perhaps a table of the p values for these? We would like to add it as supplementary tables S2 and S3

Line 218- comma after genes DONE

Line 218- neither were there … (reword) DONE

Line 222- It would be good to have in here which colour means positive? DONE

Line 237- maybe change agent for pathogen/ bacteria? DONE

Line 239-240- do these gene names need to be italicised? DONE

Line 261- please reword this as it is unclear what you are saying DONE

Line 262- within the SHV group (reword) DONE

Line 263- with SHV-28  being the most frequent one (reword) DONE

Line 279- maybe have this as humans? DONE

Line 285- you have enterobacteria with a capital and a non capital throughout, please check and be consistent DONE

Line 291- delete an DONE

Line 306- efficient at removing all … (reword) DONE

Line 311- Considering that most of the hedgehog’s (reword) DONE

Line 356- 258- please reword this as it is a bit unclear DONE

Reviewer 2 Report

The manuscript "Extended-spectrum β-lactam resistant Klebsiella pneumoniae and Escherichia coli in wild European hedgehogs (Erinaceus europeus) living in populated areas" by Garcias et al. deals with a very interesting topic and certainly in a One Health perspective the study of wild species that live in close contacts with humans is a crucial point and an useful tool to monitor the spread of resistant bacteria and AMR genes in the environment. However, there are a few points that should be revised, mainly in the Materials and Methods and Results paragraphs. Please see the list below.

Introduction

  • Line 75: I agree that for hedgehog populations the urban environment is the most favourable and the one in which they usually live but I think that in addition to the emphasis related to the close contacts to humans the possibility of acquisition from farms, stressing what is reported in line 85, should also be emphasized.

Materials and Methods

  • Lines 96-99: A map showing the sampling area, municipalities and county, the sampling sites of hedgehogs and the related human density should be added. For example Figure 3, without data of presence/absence of AMR genes, could be moved to “Results” paragraph, as over there it does not give any additional information since none of the variables had an effect on the presence/absence of genes.
  • Lines 101-108: Was age and gender also recorded at admission? These epidemiological information should be briefly introduced.
  • Lines 113-114. Why Authors decided to use these three type of agar considering that the study focused on ESBL?
  • Lines 115-116: is bacterial identification done for isolates grown on all the three agar used?
  • Lines 119-121: did Authors carry out antimicrobial susceptibility test on isolates grown on MacConkey agar + ceftriaxone?
  • Lines 142-144: several AMR genes were investigated, a table with the sequences of primers used should be added. In addition, I think the reference [18] is not the correct one, since it does not refer to resistance genes and PCR reactions.
  • Lines 157-166: a more in-depth description of the used GLMs should be added: was the dependent variable “presence/absence of any AMR gene”? What were the explanatory variables?

Results

  • Line 176: are overall 114 the faecal samples analyzed or 115, as Authors wrote in line 96? Prevalences were calculated over a total of 114 samples and also in Table 1 the total sampling is 114.
  • Lines 176-177: does this sentence mean that from each sample grown (n = 67) on MacConkey agar + ceftriaxone not a single colony was analysed?
  • Line 178: what was the prevalence of  Klebsiella pneumoniae? The finding of this pathogen is emphasized starting from the title so including its specific prevalence is necessary.
  • Line 183: what was the prevalence of AMR genes recorded in Klebsiella pneumoniae?
  • Table 1: the sum of the “number of animals” is 76. Were not 67 out of 114 samples the wild hedgehog samples that had a positive growth on MK + ceftriaxone? Otherwise, if the same animal had more than one gene, this point should be better clarified in this table because, as it is written, is confused.
  • Was the “bacterial species” included as explanatory variable in the regression model? I agree with the Authors' approach to consider as positive all the strains that had a molecular positivity for one of the AMR genes regardless of the fact that they were Klebsiella spp or E. coli or etc. However, I think that, even risking to fragment the sample, it would be interesting to carry out also specific regression models on the most prevalent strains (e.g. Klebsiella spp.) in order to investigate whether there were any association between specific strain and the epidemiological factors analyzed.
  • Line 195: I do not understand how many isolates were tested: initially 67 strains were grown on MacConkey agar + ceftriaxone. From these, 42 isolates expressed AMR genes but then Authors refer to 52 isolates (“The most common combination was blaCTX-M-15 and blaSHV-28 that was present in 21.2 % (11/52) of the isolates”) which are the ones that were also included in the antimicrobial susceptibility test. From which medium were they isolated? Were they identified by API?
  • Table 2: now the list shows 54 isolates. Again, I do not understand how many isolates were tested.
  • Lines 204-207: from these sentences it is clear that the antimicrobial susceptibility test was not carried out on isolates grown on MacConkey agar + ceftriaxone. If the manuscript is focused on ESBLs (starting with the emphasis of title), why not perform the antimicrobial susceptibility test on isolates that had an ESBL phenotype (grown on MK + ceftriaxone) and show resistances of isolates that are not necessarily ESBL?
  • Line 213: Why did you decide to include the explanatory variable “month of collection of the sample” rather than “year”?

Discussion

  • Lines 237-238: the prevalence of AMR genes of Klebsiella pneumoniae was not even mentioned in the Results paragraph. That of Klebsiella spp. was reported.
  • Line 259: I do not understand which "both environment" Authors refer to. Please rephrased this sentence.
  • In addition to the possibility of colistin (mcr genes), I think the potential risks of AMR bacteria and genes spread from farms, especially pig farms in light of the high density reported, should be included as hypothesis in this paragraph.

Minor revisions

  • Lines 176-189: genus and species of each bacterium should be written in italics.
  • Line 269: a reference supporting this statement should be added.
  • Line 354: if this is the first study on β-lactam resistant enterobacteria in wild hedgehogs, it would be better to use another verb instead of "confirm".

Author Response

Introduction

  • Line 75: I agree that for hedgehog populations the urban environment is the most favourable and the one in which they usually live but I think that in addition to the emphasis related to the close contacts to humans the possibility of acquisition from farms, stressing what is reported in line 85, should also be emphasized. This part of the introduction has changed by:

“Some of the consequences of this close hedgehog-human interaction is that people usually feed these animals and thus, increases the risk of human physical contact. Moreover, the presence of hedgehogs in other anthropogenic environment as farms or crop fields can also allow the contact with livestock.”

Materials and Methods

  • Lines 96-99: A map showing the sampling area, municipalities and county, the sampling sites of hedgehogs and the related human density should be added. For example Figure 3, without data of presence/absence of AMR genes, could be moved to “Results” paragraph, as over there it does not give any additional information since none of the variables had an effect on the presence/absence of genes. See new Figure 1.
  • Lines 101-108: Was age and gender also recorded at admission? These epidemiological information should be briefly introduced. DONE
  • Lines 113-114. Why Authors decided to use these three type of agar considering that the study focused on ESBL? We decided to use agar without ceftriaxone supplementation to have a quality control of the sample growth. For instance, to assess negative growths on supplemented agar due to the effect of the ceftriaxone activity instead of a bad quality or degradation of the sample.
  • Lines 115-116: is bacterial identification done for isolates grown on all the three agar used? Bacterial identification was just done on agar supplemented with ceftriaxone
  • Lines 119-121: did Authors carry out antimicrobial susceptibility test on isolates grown on MacConkey agar + ceftriaxone? Isolates grown on MacConkey agar with ceftriaxone were re-cultured on TSA agar and antimicrobial susceptibility test using Kirby-Bauer method was performed then.
  • Lines 142-144: several AMR genes were investigated, a table with the sequences of primers used should be added. In addition, I think the reference [18] is not the correct one, since it does not refer to resistance genes and PCR reactions. Correction done. A supplemented table S1 was included with primer sequences information.
  • Lines 157-166: a more in-depth description of the used GLMs should be added: was the dependent variable “presence/absence of any AMR gene”? What were the explanatory variables? All the statistical analyses were performed with R software, version 3.6.0 [23]. Graphics for representing antimicrobial susceptibility were built with the “AMR” package [24]. Generalised lineal models were used to test the association between the presence of AMR genes (considered as dependent variable) and different variables such as age, sex, month, population, density of the municipality and cause of admission to WRC (considered as explanatory variables).

Results

  • Line 176: are overall 114 the faecal samples analyzed or 115, as Authors wrote in line 96? Prevalences were calculated over a total of 114 samples and also in Table 1 the total sampling is 114. There are 114 fecal samples, the mistake has been corrected.
  • Lines 176-177: does this sentence mean that from each sample grown (n = 67) on MacConkey agar + ceftriaxone not a single colony was analysed? It means that 67 animals were carriers of bacteria that grew on supplemented agar. The text has been modified accordingly:

Sixty seven out of 114 wild hedgehogs analyzed (58.3%) had a positive growth on agar supplemented with ceftriaxone. Most of the animals presented pure cultures and 19 out of 67 hedgehogs presented more than one colony, identifying a total of 90 bacterial isolates. Klebsiella spp, principally Klebsiella pneumoniae (29 % out of the total bacteria), was the most frequently detected microorganism (42.2%), followed by E. coli (38.9%).

  • Line 178: what was the prevalence of Klebsiella pneumoniae? The finding of this pathogen is emphasized starting from the title so including its specific prevalence is necessary. DONE above.
  • Line 183: what was the prevalence of AMR genes recorded inKlebsiella pneumoniae?

Interestingly, a high proportion of the bacterial isolates presented more than one AMR gene, being especially remarkable within the genus Klebsiella, in which 67.7 % (21/31) of isolates presented two or more β-lactamase genes. The most common combination was blaCTX-M-15 and blaSHV-28 that was present in 21.2 % (11/52) of the isolates. Moreover, it’s especially remarkable the high prevalence of K. pneumoniae (84.6 %) carrying AMR genes, detecting up to 4-5 genes in some isolates (Table 2).

  • Table 1: the sum of the “number of animals” is 76. Were not 67 out of 114 samples the wild hedgehog samples that had a positive growth on MK + ceftriaxone? Otherwise, if the same animal had more than one gene, this point should be better clarified in this table because, as it is written, is confused. There are animals that carried more than a gene, so the sum is bigger. It was explained better above.
  • Was the “bacterial species” included as explanatory variable in the regression model? I agree with the Authors' approach to consider as positive all the strains that had a molecular positivity for one of the AMR genes regardless of the fact that they were Klebsiellaspp or  coli or etc. However, I think that, even risking to fragment the sample, it would be interesting to carry out also specific regression models on the most prevalent strains (e.g. Klebsiella spp.) in order to investigate whether there were any association between specific strain and the epidemiological factors analyzed.  DONE and the results were not significant. See Table below:

Odd-ratios and p-values of bivariate generalized linear models for specific Klebsiella data

OR

OR CI 95%

p-value

Age group (Model 1)

Adult

-

-

-

Juvenile

3.2

(0.4, 27.4)

0.248

Sex (Model 2)

Female

-

-

-

Male

1.1

(0.2, 9.6)

0.905

Month (Model 3)

October

-

-

-

April

1

(0, Inf)

1

May

1

(0, Inf)

1

June

9.5x 10^-9

(0, Inf)

0.997

July

4.7x 10^-7

(0, Inf)

0.997

August

9.5x 10^-9

(0, Inf)

0.997

September

1

(0, Inf)

1

November

1.6x 10^-8

(0, Inf)

0.997

December

1

(0, Inf)

1

Cause of admission (Model 4)

Fortuitous

-

-

-

Accident

0.5

(0.001, 17.5)

0.676

Breeding

2.25

(0.007, 67.6)

0.598

Trauma

289.1

(0, Inf)

1

Weakness

1.25

(0.004, 39)

0.887

Other

1

(0.003, 31.8)

1

Population density (Model 5)

>1000 ha/m2

-

-

-

1000-2250 ha/m2

350

(0, Inf)

0.996

2250-5000 ha/m2

1.7

(0.1, 41.4)

0.682

>5000 ha/m2

3.4

(0.3, 78.6)

0.33

If the Reviewer considers the table relevant we can provide it as a suplemmentary table S4.

  • Line 195: I do not understand how many isolates were tested: initially 67 strains were grown on MacConkey agar + ceftriaxone. From these, 42 isolates expressed AMR genes but then Authors refer to 52 isolates (“The most common combination was blaCTX-M-15 and blaSHV-28 that was present in 21.2 % (11/52) of the isolates”) which are the ones that were also included in the antimicrobial susceptibility test. From which medium were they isolated? Were they identified by API? We have tried to clarify adding a paragraph on discussion. The fact is that 67 animals have growth in supplemented agar and 90 bacteria were isolated from there. However, when we performed molecular analysis we saw that there was 42 animals with AMR genes with 52 genes.
  • Table 2: now the list shows 54 isolates. Again, I do not understand how many isolates were tested. In Table 2, there are 52 genes and the discussion is more detailed:

Sixty seven out of 114 animals (57.8 %) have growth in ceftriaxone supplemented agar but AMR genes were detected in 42 of them (36.8 %). As regards the bacterial isolates, a total of 90 bacterial isolates were identified from those 67 animals, detecting the presence of AMR genes in 52 isolates. Thus, in line with other studies [28,29], we observed that some of the AMRB did not have any of the AMR genes. This could be explain by the possible existence of other mechanisms of resistance such as chromosomal mechanisms that were not included as a target of this study.

  • Lines 204-207: from these sentences it is clear that the antimicrobial susceptibility test was not carried out on isolates grown on MacConkey agar + ceftriaxone. If the manuscript is focused on ESBLs (starting with the emphasis of title), why not perform the antimicrobial susceptibility test on isolates that had an ESBL phenotype (grown on MK + ceftriaxone) and show resistances of isolates that are not necessarily ESBL? The antimicrobial susceptibility test (AST) were just performed on isolates grown on MacConkey agar + ceftriaxone, in order to focus on ESBLs. And this AMR genes were found on these isolates. However, AST was done after quarantine, so that means that isolates were frozen at -80ºC and this could affect to the viability of some of the strains and lose some of the genes that confer resistance. However, we don’t have a certainty of that and we don’t have any other explanation.
  • Line 213: Why did you decide to include the explanatory variable “month of collection of the sample” rather than “year”? Using year was considered but the problem was that the most of the samples were collected during 2019 and 2018 and in the other years the sample was small so we considered that could create misconceptions. Using month, we want to assess the influence of seasonality on the presentation of AMR genes, but any relationship was found with this data.

Discussion

  • Lines 237-238: the prevalence of AMR genes of Klebsiella pneumoniaewas not even mentioned in the Results paragraph. That of Klebsiella  was reported. Now mentioned
  • Line 259: I do not understand which "both environment" Authors refer to. Please rephrased this sentence. DONE
  • In addition to the possibility of colistin (mcr genes), I think the potential risks of AMR bacteria and genes spread from farms, especially pig farms in light of the high density reported, should be included as hypothesis in this paragraph. DONE

In this study, none of the tested animals presented colistin (mcr-1 to -5) resistance genes, but this is an expected finding since most of the tested animals came from urbanized areas with a low risk of contact with residues from pig production. However, the inclusion of a larger sample size with hedgehogs from regions with a high swine population density, like Osona, in the analysis, could help to improve the odd of detecting these colistin resistance genes.

Minor revisions

  • Lines 176-189: genus and species of each bacterium should be written in italics. DONE
  • Line 269: a reference supporting this statement should be added. DONE
  • Line 354: if this is the first study on β-lactam resistant enterobacteria in wild hedgehogs, it would be better to use another verb instead of "confirm". changed by “reveals”